# Uniform-Density Micro-Textured Ball-End Milling Cutter Model Based on Fractal and Uniform Distribution Theory

**Pei Han, Xin Tong \*, Shucai Yang and Xiyue Wang**

Key Laboratory of Advanced Manufacturing and Intelligent Technology, Ministry of Education, School of Mechanical and Power Engineering, Harbin University of Science and Technology, Harbin 150080, China; 2110103011@stu.hrbust.edu.cn (P.H.); 18729280819@163.com (S.Y.); m13555330606@163.com (X.W.)
\* Correspondence: tongxin@hrbust.edu.cn

**Abstract:** At present, for micro-textured tools, the determination of the micro-texture placement area depends on the derivation of the cutting geometric model. The micro-texture distribution form applies geometric methods, and the research methods and accuracy are limited. Therefore, in this paper, the ball-end milling cutter is taken as the research object. Based on fractal theory, the morphology of the tool before and after wear is compared to determine the tool–chip contact area. The uniform-density micro-texture distribution model is established using the uniform distribution point theorem, and the synergistic mechanism of the edge and the micro-texture is revealed. The strength of the micro-textured tool with uniform density under the action of the edge is studied by simulation. Finally, the determination of the tool–chip contact area and the establishment of a uniform-density micro-texture model is realized. It is proved that the synergistic effect of the cutting edge and the micro-texture has a positive effect on the milling behavior of the tool. When comparing the non-edge and non-texture tools with the cutting-edge tools, the maximum strain and stress of the cutting-edge micro-textured tools increased by 12% and 30%, and 30% and 20%, respectively, without affecting the normal use of the tool. This research provides a new method for the design of micro-textured tools.

**Keywords:** tool–chip contact area; micro-texture; fractal theory; blunt round edge; distribution density function; tool strength

## 1. Introduction

Due to the strong chemical activity of titanium alloys, problems such as high milling force, high milling temperature and serious tool wear can easily arise during milling [1]. Studies have shown that the placement of micro-texture can effectively improve the anti-wear and anti-friction performance of the tool and improve the efficiency and quality of cutting. At present, most of the research on micro-textured tools is related to the following aspects: (1) the cutting performance of micro-textured tools; (2) the placement area and distribution form of micro-texture; (3) optimization of micro-texture parameters; (4) the preparation process for tool micro-texture; (5) the synergistic effect of the micro-texture and the cutting edge; and (6) the combination of micro-texture and coatings [2–4]. A study of the placement area and distribution form of micro-texture can serve as the basis for discussing the effectiveness of micro-texture tools. Rational placement and design are the premise for the role played by the micro-texture. In precision machining, a shallow cutting depth and a low feed rate, as well as a high cutting speed, are usually chosen to ensure precision. The main cutting is completed by the cutting edge and the tool surface near the edge. Severe wear usually occurs at the cutting edge and in nearby areas, resulting in poor surface quality of the workpiece. Therefore, passivation treatment of the tool edge can effectively improve the tool life and the stability of the cutting process, while also improving the cutting performance of the tool. It is necessary to study the micro-textured tool under the action of the cutting edge.

Many scholars have conducted research on the relevance of micro-texture in recent years. Veiga F. [5] and He [6] jointly modeled the temperature of the tool–chip contact area during the cutting process. The former proposed a new method that takes into account the temperature-dependent thermal conductivity of the material in order to estimate the heat transfer ratio on the secondary heat source. The latter developed a mathematical model of the temperature field in the tool chip region based on the cutting relationship. Liu [7] studied the friction coefficient of the tool–chip contact area during oblique cutting using AdvantEdge (metal cutting simulation software). Yang et al. [8] determined the contact area between the tool and the chip by measuring the milling force during the milling process with a ball-end milling cutter, and established a milling model using analytical methods. Ye [9] obtained the milling force interval algorithm for ball-end milling cutters based on the cutting situation of a neutral milling cutter during the actual machining process. Qi [10], Li [11], Wu [12], and Patel [13] all found that embedding micro-textures in the tool–chip contact area can significantly reduce the cutting force required during the cutting process, significantly reduce the cutting temperature, and improve the machining quality of the workpiece. Kishawy [14] proposed an analytical model based on Oxley (Oxley's theory is a chip formation model based on slip line field analysis and strain rate analysis of the plastic flow field) to optimize the design of micro-texture cutting tools with the aim of suppressing the occurrence of derivative cutting. Through orthogonal cutting experiments on AISI 1045 steel pipes, the optimal micro-texture design for eliminating derivative cutting and reducing cutting force was obtained. Fan et al. [15] processed five types of micro-texture on the rake face of the PCBN tools. The effects of different microstructures on the cutting force, cutting temperature, micro-morphology of serrated chips, tool wear, and surface roughness were studied. The results showed that, compared with non-textured tools, micro-textured tools demonstrated reduced cutting force, but the generated cutting force changed more significantly during the cutting process. Micro-texture can reduce the temperature in the tool–chip contact area, change the temperature distribution on the rake face, and reduce the maximum temperature at the edge. Machined surfaces with elliptical and wavy grooves have good wear resistance and surface quality. The synergistic effect of the micro-texture and the cutting edge may also have a further impact on the milling process. Tong [16] established a finite element analysis model for milling titanium alloy with micro-textured ball-end milling cutters under different blade forms. Through simulation analysis, the influence of the micro-texture parameters and blade parameters on the cutting performance of the tool was studied. Yang [17] and Zhang [18] studied the performance of micro-textured ball-end milling cutters with different blade types for milling titanium alloys. They found that the presence of the blade improved the structural strength and impact resistance of the cutting edge, and the synergistic effect of the two significantly improved the cutting performance of the tool. Gurgen et al. [19] used three different multi-objective models to determine the optimal cutting and geometric parameters for SPRT (self-propelled rotary tool) turning operations, and studied the effects of feed speed and spindle speed on the surface roughness and metal removal rate. Orak et al. [20] determined the optimal parameters for chatter during the turning process and developed a hybrid decision algorithm composed of an artificial neural network, using the TOPSIS method to optimize machining parameters.

In summary, it is not difficult to find that the micro-texture placement of the tool is usually in the tool–chip contact area, and the discussion of the tool–chip contact relationship is mostly based on theoretical analysis. In real-world situations, because the cutting parameters and milling state will change with changes in machine tool performance, the possibility of determining the tool–chip contact area is limited. Fractal theory can be used to describe and study objective things from the perspective of fractional dimensions and mathematical methods, providing a new method for the determination of the tool–chip contact area. In this paper, the tool–chip contact area is determined based on fractal theory, and the uniform-distribution density function of the tool micro-texture is established using the random point generation theorem of uniform distributions in elliptical regions. A three-

dimensional milling model is used to analyze the influence of the micro-texture and the edge on the tool and its effect. The influence of the micro-texture and the edge on tool strength is studied using the finite element method.

## 2. Determination of Cutter–Chip Contact Area of a Ball-End Milling Cutter Based on Fractal Theory

### 2.1. Materials and Methods

Hard-alloy ball-end milling cutters are widely used for machining difficult-to-machine materials due to their excellent wear resistance and mechanical properties. Titanium alloy is one of the typical representatives of difficult-to-machine materials. Titanium alloy Ti6Al4V, as an ideal lightweight material, has excellent properties, such as corrosion resistance, high temperature resistance, and high specific strength. It is widely used in aerospace, national defense technology, and other fields, and has become one of the indispensable metal materials in the mechanical manufacturing industry [21,22]. Therefore, it is necessary to study the processing behavior of titanium alloy milling with hard-alloy ball-end milling cutters.

The selection of cutting parameters is crucial when using ball-end milling cutters for the precision milling of titanium alloys. Cutting parameters can affect the tool workpiece contact relationship and chip removal, ultimately affecting the stability of the milling system and the machining quality of the workpiece. In order to ensure the performance of ball-end milling cutters, with reference to the CNC tool selection guide and the team's previous research results [23,24], the axial cutting depth range is set to [0.3, 0.5 mm], the feed rate range is [0.06, 1 mm/r], and the cutting speed is [120, 160 m/min]. Due to the characteristic of "balanced dispersion, orderliness and comparability" in orthogonal tables, a few representative experimental conditions can be selected within the inspection range to achieve balanced sampling [25]. Therefore, a three-factor and three-level orthogonal experiment is designed to explore the tool–chip contact area under the conditions of the precision machining of titanium alloy with a ball-end milling cutter. The orthogonal table of the design of experiments is shown in Table 1.

**Table 1.** Orthogonal test table of cutting parameters.

| Test Number | Axial Cutting Depth $a_p$ (mm) | Feed Rate $f$ (mm/r) | Cutting Speed $v$ (m/min) |
|:---:|:---:|:---:|:---:|
| 1 | 0.3 | 0.06 | 120 |
| 2 | 0.3 | 0.08 | 140 |
| 3 | 0.3 | 0.1 | 160 |
| 4 | 0.4 | 0.06 | 140 |
| 5 | 0.4 | 0.08 | 160 |
| 6 | 0.4 | 0.1 | 120 |
| 7 | 0.5 | 0.06 | 160 |
| 8 | 0.5 | 0.08 | 120 |
| 9 | 0.5 | 0.1 | 140 |

The cutter blade model is BNM-200 (BNM-200 produced by Xiamen Golden Egret Special Alloy Co., Ltd., Xiamen, China), the tool material is cemented carbide, the brand is YG8 (YG8 tool produced by Xiamen Golden Egret Special Alloy Co., Ltd., Xiamen, China), there is no coating, the rake angle is zero, and the toolholder is cemented carbide with a size of $\Phi 20 \times 141$ mm, as shown in Figure 1a. The titanium alloy material with a workpiece grade of TC4 used in the test is shown in Figure 1b, and the relevant physical and chemical parameters of these materials are shown in Figure 2. A VDL-1000 E (VDL-1000E produced by Dalian Machine Tool Factory, Dalian, China) three-axis vertical milling machine was selected as the machine tool, and is shown in Figure 1c. The clamping method of the titanium alloy workpiece includes an angle between the surface to be machined and the horizontal plane of the machine tool guide rail of 15°, as shown in Figure 1d. Considering the advantages of down milling, such as stable milling, high surface quality, and slow tool

wear, the down milling method was chosen. Before and after the test, the surface morphology of the tool was photographed using a super depth of field microscope (produced by Pinzhi Chuangsi Precision Instrument Co., Ltd., Beijing, China).

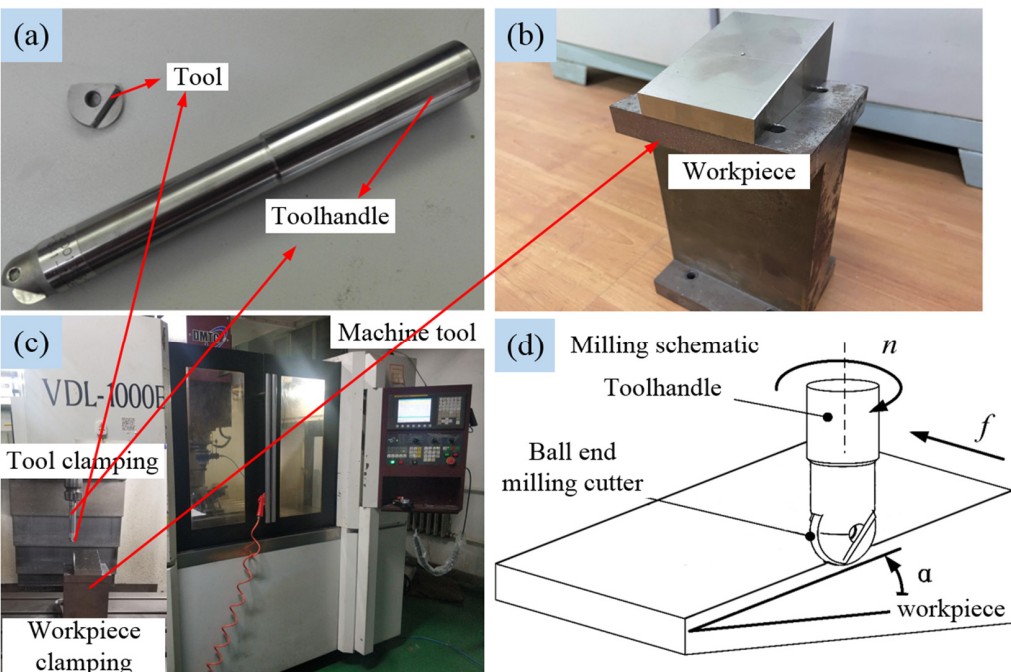

**Figure 1.** Experimental materials. (**a**) Tool, (**b**) Workpiece, (**c**) Testing site, (**d**) Milling method. *n*: Tool rotation direction, *f*: feed direction.

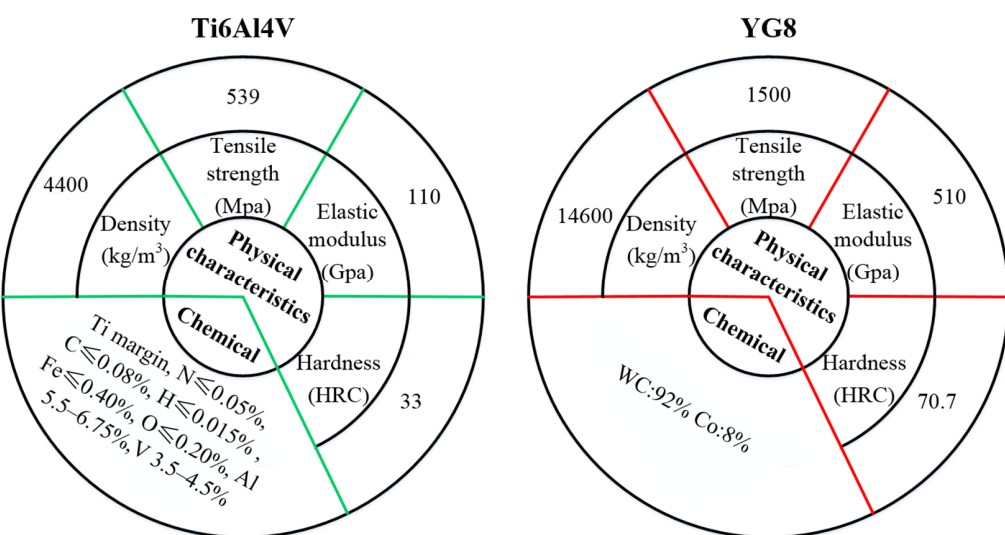

**Figure 2.** Physical and chemical parameters of materials.

### 2.2. Tool Wear Results

During the research process, various graphical and image results provide the most intuitive form for recording physical information related to the research object. These graphic images contain a wealth of physical information about the research object [26]. In the case of the tool wear analysis, capturing tool wear images is the most direct approach for further analysis. Graphics are typically processed in the form of digital images or can be converted into digital images. Digital images exhibit characteristics of quantization and discretization. Moreover, research has demonstrated that fractal shapes and behaviors are universal, and occur in both the natural world and in the domain of human subjective

thinking. Fractals exhibit fundamental properties of self-similarity, scale-free behavior, and self-radiation [27]. Within the field of mechanical science, phenomena such as mechanical vibrations, surface morphology, fractal springs, friction and wear, and fault diagnosis exhibit fractal characteristics. For instance, in the context of milling titanium alloy using ball-end milling cutters, tool–chip and tool-workpiece friction lead to tool wear. Analyzing tool wear images, the distribution relationship between pixel positions and image intensity within the image area can accurately reflect the true surface morphology. By representing the grayscale value of pixels as the height of a plane, the image's grayscale values effectively reveal the surface morphology at different heights, thereby reflecting the 'roughness' of the real surface [28].

To examine the self-similarity of the tool image, the same position on the tool is photographed using super depth of field, and repeatedly enlarged. The results are presented in Figure 3. It is evident that the structures obtained from each level of magnification display similarities to the overall structure. Thus, characterizing the degree of tool wear through fractal theory is a feasible approach. The tool wear area is the contact area between the tool and the chip.

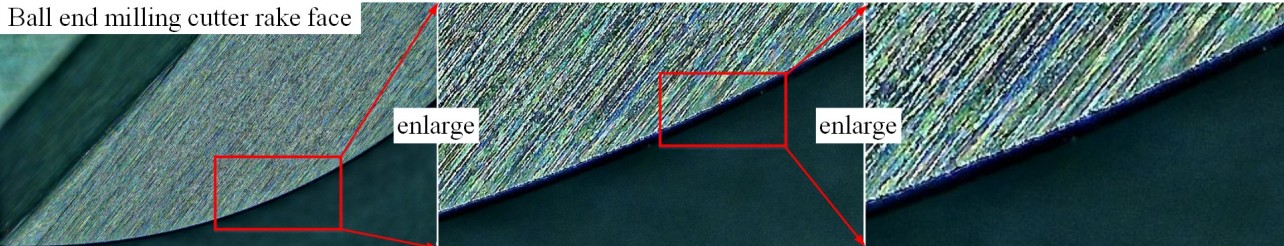

**Figure 3.** Similarity test.

There are differences in surface morphology before and after tool use. In order to eliminate this error, the difference between the fractal dimensions before and after the use of the tool is obtained as the tool wear result, and is then further analyzed. The processing is shown in Figure 4. The sample tool 1 is taken as an example in order to explain the results. Using MATLAB programming, the division of the tool area and the calculation of the box dimensions are realized. In the initial calculation, the picture is divided into 100 equal parts to determine the box dimensions of each part. It should be noted that when the rake face of the tool is photographed, other areas of the tool will be photographed, which will interfere with the calculation results. Therefore, the results of the interference area need to be eliminated when calculating the average. When the box dimensions are larger [29], the rougher the tool surface, the more serious the wear. The initially determined tool–chip contact area is shown in Figure 5.

Due to the size of the first segmented image being too large, the segmented image was further refined. The box dimensions of the image are solved to obtain the final tool–chip contact area, as shown in Figure 6. The final tool–chip contact area is shown in Figure 6. On the basis of the calculation results of the box dimensions, the area of severe tool wear is also identified, as indicated by the yellow area in Figure 7. The main form of wear is adhesive wear.

In order to establish the micro-texture distribution model of the ball-end milling cutter, a rectangular plane coordinate system is established with the tool tip as the origin, the cutting-edge tangent as the X-axis direction, the vertical cutting direction as the Y-axis direction, and the image segmentation size as the coordinate scale. The curve is fitted to solve the tool–chip contact area equation. A schematic diagram is shown in Figure 8.

According to Figure 8, the curve of the tool–chip contact area is mainly composed of four curves. Curves $y_1$ and $y_2$ are $x_1 = 384$, $x_2 = 1828$, curve $y_3$ is a circle passing through the origin (0,0) and the point (384,27), curve $y_4$ is a curve surrounded by the above-marked points. Data fitting is performed for curve $y_4$, and the fitting equation is

shown in Equation (1). Significance test: $R_2$ is 0.963, $F$ statistic is 209, $p$ value is $2.5 \times 10^{-27}$. It is proved that the fitting equation is significant, and the results are shown in Figure 9.

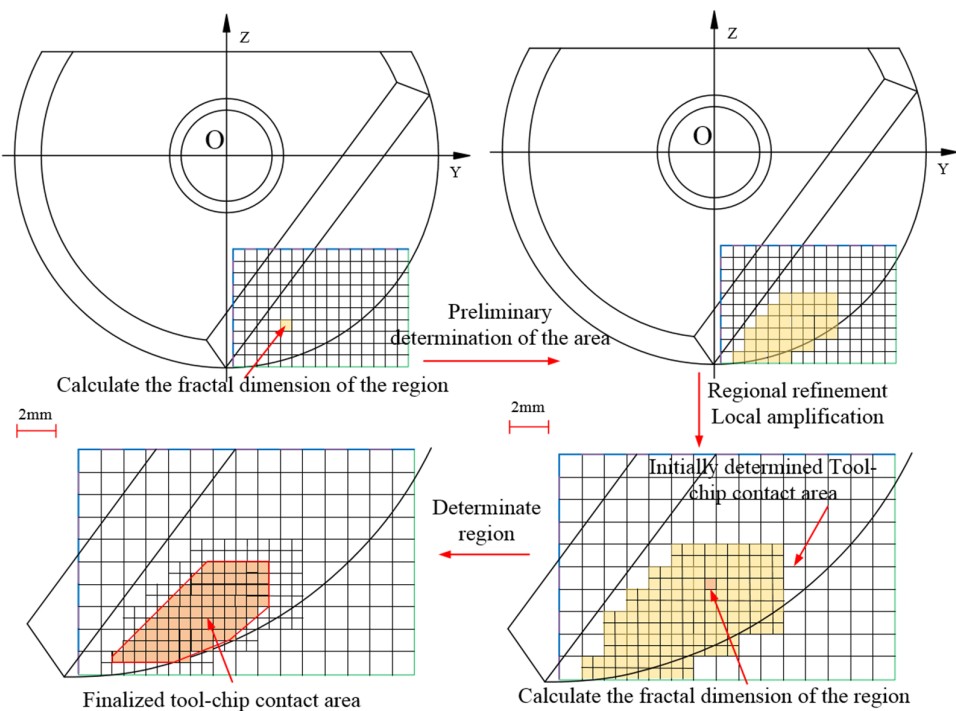

**Figure 4.** Specific processing flow chart.

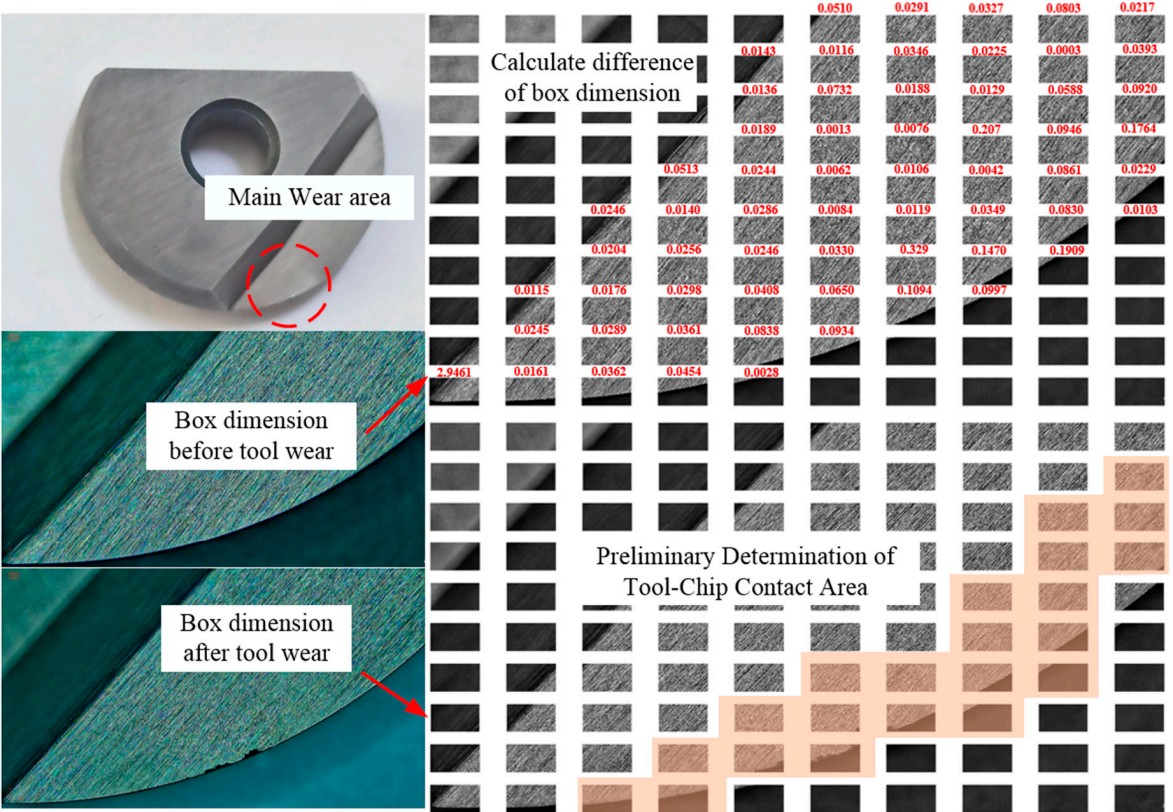

**Figure 5.** Sample tool 1—the preliminarily determined tool–chip contact area.

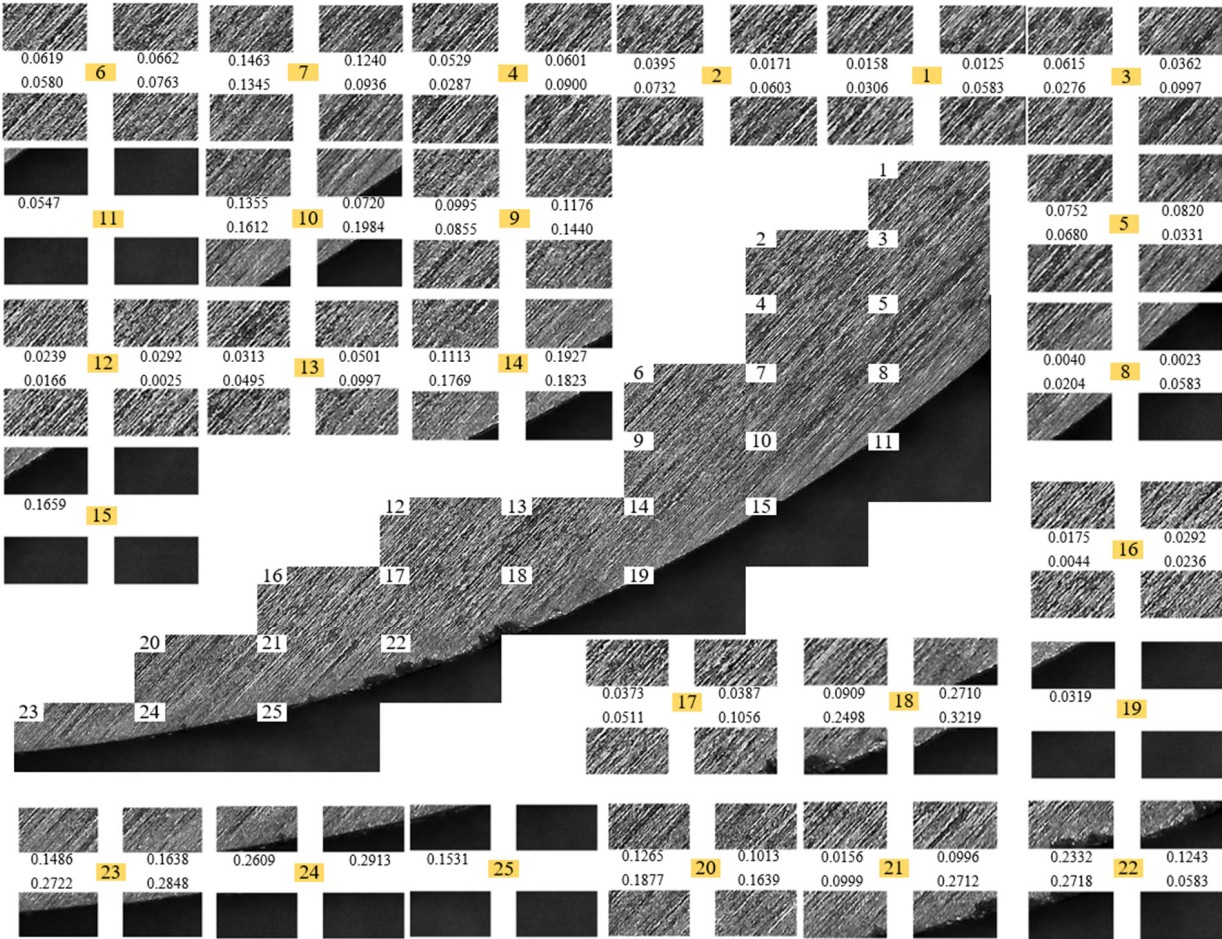

**Figure 6.** Box dimension calculation results—the refined tool–chip contact area.

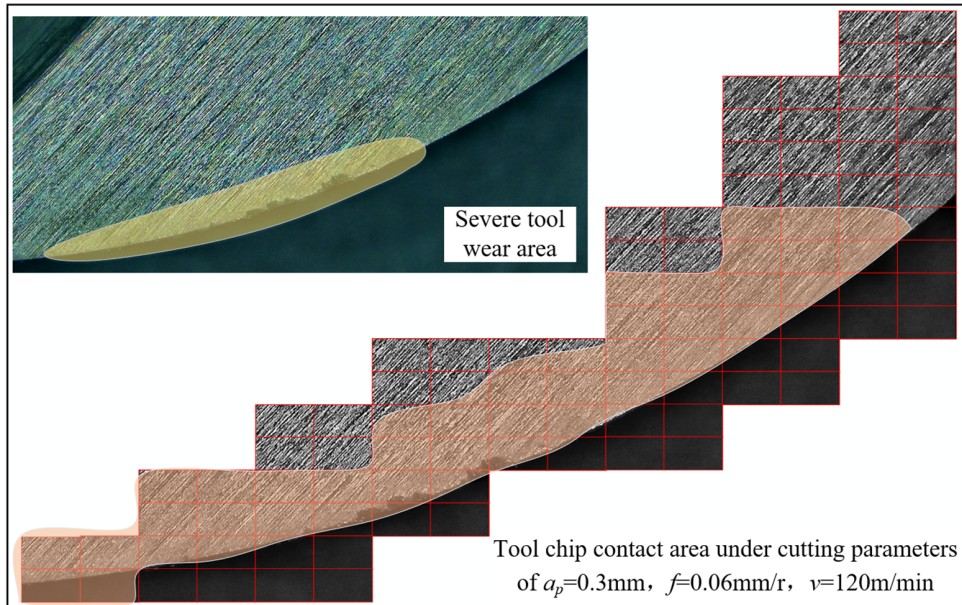

**Figure 7.** Sample tool 1 tool–chip contact area.

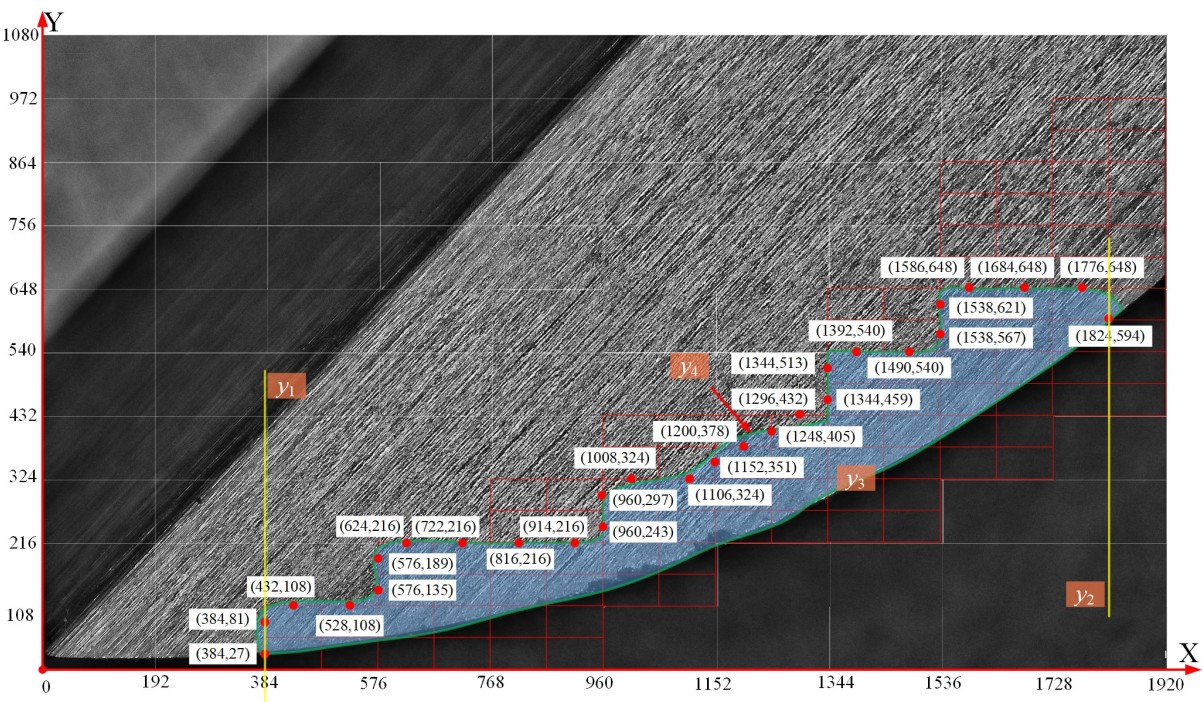

**Figure 8.** Sample tool 1 cutter–chip contact area coordinate system.

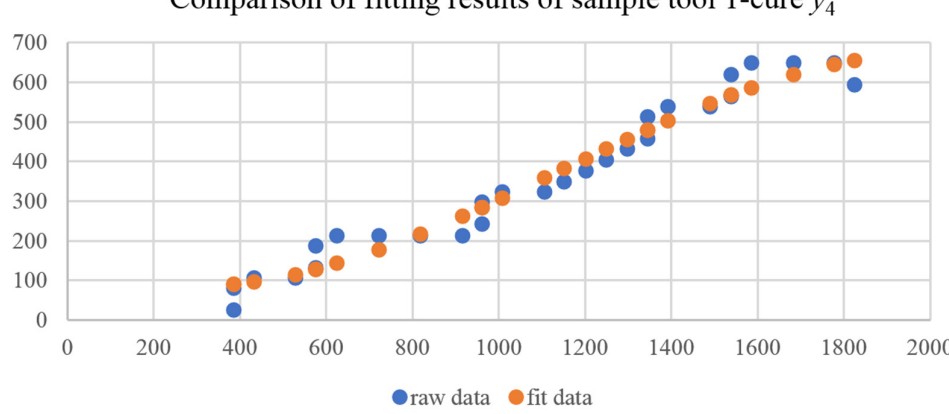

**Figure 9.** Comparison of curve $y_4$ fitting results.

Finally, the tool–chip contact area equation is as follows:

$$
\begin{cases}
x_1 = 384 \\
x_2 = 1828 \\
y_3 = 146.53 - 0.42142x_3 + 0.0008173x_3^2 - 2.3728 \times 10^{-7}x_3^3 \ (384 < x_3 < 1828) \\
2744.166^2 = x_4^2 + (y_4 - 2744.166)^2 \ (0 < x_4 < 1920)
\end{cases}
\tag{1}
$$

In order to test the accuracy of the equation, the curve equation is drawn using MATLAB. The results are shown in Figure 10. It can be seen that the curve equation results are in good agreement with the original results. The effectiveness of this method of analysis is also proved.

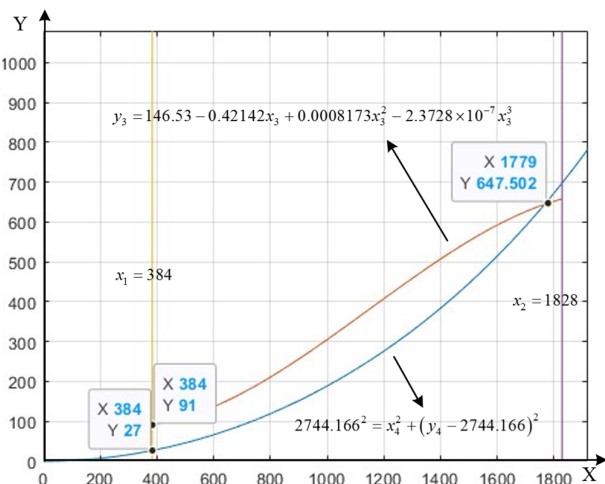

**Figure 10.** Curve equation simulation results.

The same treatment was performed on the nine tools in the experiment to obtain the corresponding tool–chip contact area equation. The equation solved is as follows (the arc contour curve equation of the tool is the same as that of tool 1, and the following equation is not given):

$$
\begin{cases}
x_1 = 576 \\
x_2 = 1920 \\
y_3 = 365.43 - 1.1201x_3 + 0.0014348x_3^2 \\
-4.0456 \times 10^{-7}x_3^3 \ (576 < x_3 < 1920)
\end{cases} \text{(Tool 2)}
\qquad
\begin{cases}
x_1 = 769 \\
x_2 = 1824 \\
y_3 = -333.74 + 0.70543x_3 - 5.4572 \\
\times 10^{-5}x_3^2 \ (480 < x_3 < 1824)
\end{cases} \text{(Tool 3)}
$$

$$
\begin{cases}
x_1 = 480 \\
x_2 = 1824 \\
y_3 = -483.73 + 1.6346x_3 - 0.0010812x_3^2 \\
+3.118 \times 10^{-7}x_3^3 \ (480 < x_3 < 1824)
\end{cases} \text{(Tool 4)}
\qquad
\begin{cases}
x_1 = 480 \\
x_2 = 1824 \\
y_3 = 75.955 - 0.37237x_3 + 0.0010445x_3^2 \\
-3.7579 \times 10^{-7}x_3^3 \ (480 < x_3 < 1824)
\end{cases} \text{(Tool 5)}
$$

$$
\begin{cases}
x_1 = 576 \\
x_2 = 1728 \\
y_3 = -313.36 + 0.84589x_3 - 0.00014909 \\
x_3^2 \ (576 < x_3 < 1728)
\end{cases} \text{(Tool 6)}
\qquad
\begin{cases}
x_1 = 480 \\
x_2 = 1632 \\
y_3 = 164.56 - 0.55989x_3 + 0.0013015x_3^2 \\
-4.9645 \times 10^{-7}x_3^3 \ (480 < x_3 < 1632)
\end{cases} \text{(Tool 7)}
$$

$$
\begin{cases}
x_1 = 576 \\
x_2 = 1728 \\
y_3 = -426.12 + 1.1001x_3 - 0.00025631 \\
x_3^2 \ (576 < x_3 < 1728)
\end{cases} \text{(Tool 8)}
\qquad
\begin{cases}
x_1 = 384 \\
x_2 = 1728 \\
y_3 = -57.561 + 0.15321x_3 + 0.00067185x_3^2 \\
-2.9824 \times 10^{-7}x_3^3 \ (384 < x_3 < 1728)
\end{cases} \text{(Tool 9)}
$$

$$(2)$$

Similarly, the curve equation is drawn for each tool, as shown in Figure 11. It is not difficult to see that, due to differences in cutting parameters, the wear areas of each tool are not the same. With increasing cutting parameters, the tool wear area expands.

In order to ensure that the micro-textured tool is able to give full play to the anti-wear and anti-friction effect when finishing titanium alloy, the union of the tool–chip contact area is selected for micro-texture placement under different cutting parameters. The results of the determination of the tool–chip contact area after union are shown in Figure 12a. Then, the curve formed by the union is fit. The results are determined using Equation (3). The final tool–chip contact area is shown in Figure 12b.

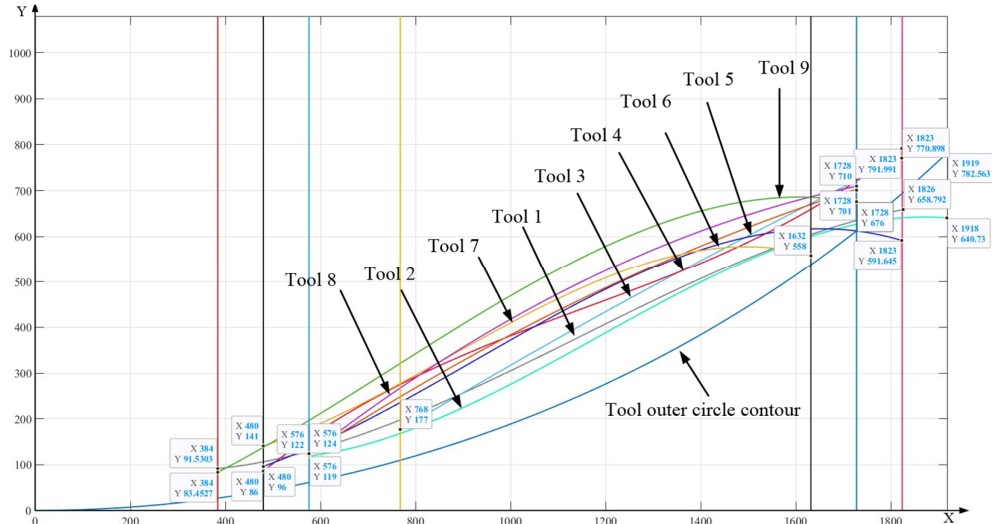

**Figure 11.** Schematic diagram of tool–chip contact area under different cutting parameters.

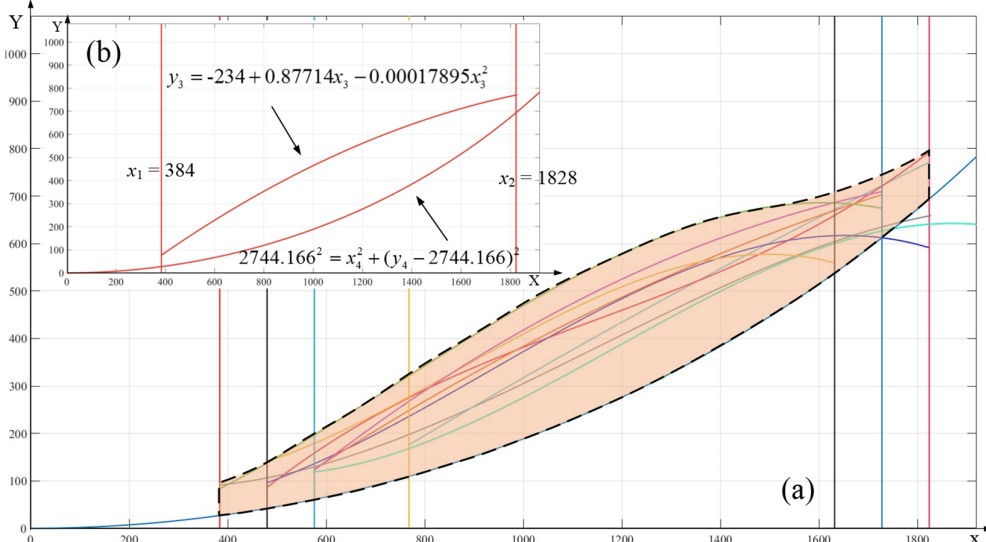

**Figure 12.** Final tool–chip contact area: (**a**) Tool-chip contact area before fitting; (**b**) Tool-chip contact area after fitting.

In order to verify the accuracy of the tool–chip contact area (as shown in Figure 12a) derived based on fractal theory, it is compared with the tool–chip contact area (as shown in Figure 13b) derived by the team based on the geometric modeling of cutting and chip curl theory in the earlier stage [30]. In combination with the actual wear image, it can be found that the former is closer to the actual wear results, indicating that it is more effective and accurate.

$$
\begin{cases}
x_1 = 384 \\
x_2 = 1828 \\
y_3 = -234 + 0.87714x_3 - 0.00017895x_3^2 \ (384 < x_3 < 1828) \\
2744.166^2 = x_4^2 + (y_4 - 2744.166)^2 \ (0 < x_4 < 1920)
\end{cases}
\tag{3}
$$

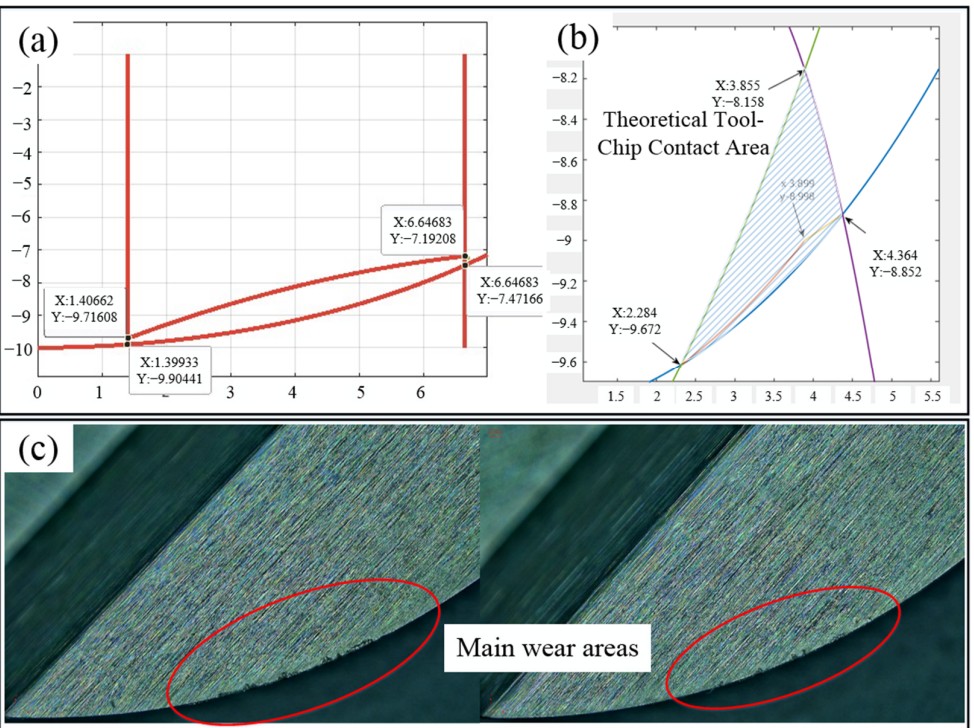

**Figure 13.** Comparison of results for tool–chip contact area. (**a**) tool–chip contact area based on fractal theory; (**b**) tool–chip contact area based on chip curling theory; (**c**) actual wear area.

## 3. Distribution Model for a Micro-Texture Ball-End Milling Cutter Based on the Distribution Density Function

Because the geometric contour of the ball-end milling cutter is circular, according to the random point generation theorem of uniform distribution in the elliptical region [31], when $a = b$, the elliptical region becomes a circular region. Then, the distribution density function in the ball-end milling cutter area can be obtained.

$$f_R(r) = \begin{cases} \frac{2r}{r_2{}^2}, & r_1 \leq r \leq r_2 \\ 0, & \text{other} \end{cases} \tag{4}$$

$$f_\Theta(\theta) = \begin{cases} \frac{1}{2\pi}, & \theta_1 \leq \theta \leq \theta_2 \\ 0, & \text{other} \end{cases} \tag{5}$$

In the formula, $r_1$ and $r_2$ are the upper and lower limits of the circle radius range; if $r_1$ is not 0, then the circular region is characterized. $\theta_1$ and $\theta_2$ are the upper and lower limits of the arc angle range; if $\theta_1$ is not 0 and $\theta_2$ is not $2\pi$, then the sector region is characterized. In the previous study, it was found that the first row of micro-looms needs to be placed at a certain distance from the cutting edge to ensure that the strength of the cutting edge is not affected. In this case, $r_2 = r_{\text{ball}} - l_1$. $l_1$ is the distance of the first row of micro-looms from the cutting edge. This results in an algorithm for the uniform distribution of micro-looms in a ball-end mill [32]:

(1) Draw the cutter–chip contact area of the ball-end milling cutter according to the curve equation.

(2) Based on the circular arc profile of the ball-end milling cutter, the circle is divided into $m_{\text{total}} = 2\pi r/l_3$ in the circumferential direction and $n_{\text{total}} = r - l/l_4$ along the radius direction, where $l_3$ is the distance between each column of micro-textures, and $l_4$ is the distance between each row of micro-textures. Set $l_3 = l_4$.

(3) Separately and independently generate U (0,1) uniform random numbers $\eta^{(1)}$ ($i = 1, 2, \dots m_{\text{total}}$) and $\eta^{(2)}$ ($j = 1, 2, \dots n_{\text{total}}$).

(4)   $\{\Theta\}$ is obtained from $\Theta = 2\pi\eta^{(1)}$, which is a random number uniformly distributed on $[0, 2\pi]$.

(5)   $\{R_j\}$ is obtained from $R_j = r_2\sqrt{\eta_j^{(2)}}$, and is a random number with a density function of $f_R(r)$.

(6)   Change $X = R_j\cos\Theta$, $Y = R_j\sin\Theta$ to form the corresponding coordinate points.

(7)   Limit the range of generating uniform random points.

The parameters in the simulation process include micro-pit diameter $D = 0.05$ mm; micro-pit distance from the cutting edge $l_1 = 0.11$ mm; micro-pit spacing $l_3 = 0.15$ mm. These parameters are input into MATLAB, and the micro-texture distribution map is then obtained according to the density distribution function.

Figure 14 presents the specific distribution of micro-texture. The specifically prepared position for the micro-texture can be determined from the diagram. From the simulation results, it can be observed that the distribution of micro-texture conforms to the uniform density distribution, and the area prepared for the micro-texture is further determined, providing a basis for the subsequent laser preparation for uniform-distribution micro-texture testing.

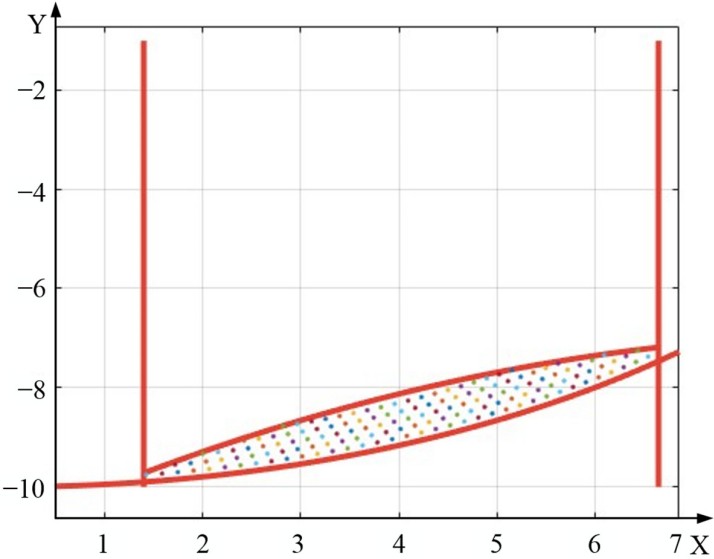

**Figure 14.** Micro-texture distribution model.

## 4. Mechanism Analysis of Uniform-Distribution-Density Micro-Texture Ball-End Milling Cutter under the Action of a Cutting Edge

The cutting phenomenon in the actual cutting process consists entirely of three-dimensional cutting. According to the three-dimensional cutting geometry model proposed by Koji Ono [33], the influence of the cutting edge and micro-texture on cutting force and cutting stress is analyzed. Through the geometric relationship shown in Figure 15, the shear force and normal force of the shear surface can be obtained.

$$\begin{cases} F_s = \tau_s \cdot A_s = bt_1 \cdot \tau_s / \sin\phi_n \cos i \\ F_n = \sigma_s \cdot A_s = bt_1 \cdot \sigma_s / \sin\phi_n \cos i \end{cases} \tag{6}$$

In the formula, $A_s$ is the area of the shear surface, $b$ is the cutting width, $t_1 = OC\sin\phi_n$ is the cutting depth, $i$ refers to the state of the tool rotation $i$ angle around the $Z$ axis in binary cutting, $\phi_n$ is the normal shear angle, $\tau_s$ is the shear stress on the shear surface, and $\sigma_s$ is the normal stress on the shear surface.

The friction force *F* between the rake face and the chip and the normal force *N* of the rake face can also be calculated as follows:

$$\begin{cases} F = \tau_t \cdot A_q = bl \cdot \tau_t \\ N = \sigma_t \cdot A_q = bl \cdot \sigma_t \end{cases} \quad (7)$$

In the formula, $A_q$ is the actual contact area between the tool and the chip, *b* is the cutting width, *l* is the contact length between the tool and the chip, $\tau_t$ is the shear stress on the rake face, and $\sigma_t$ is the normal stress on the rake face.

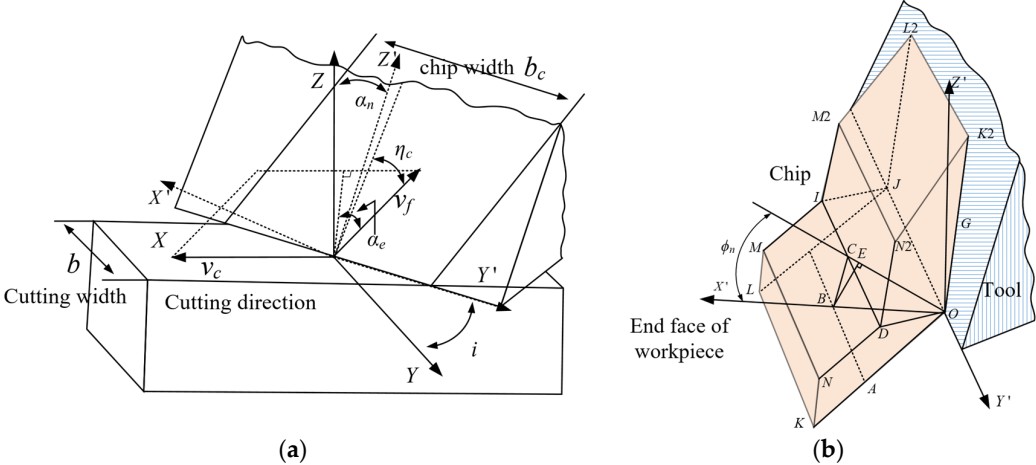

(a)                                        (b)

**Figure 15.** Three-element cutting: (**a**) three-dimensional cutting coordinate relationships; (**b**) three-dimensional cutting geometric relationships.

When the cutting edge is very sharp, wear or fracture will occur due to the concentration of stress during cutting. Therefore, the cutting edge needs to have some rounded corners. Therefore, the size of the blunt edge is the key factor in determining the cutting effect. On this basis, a mathematical model of milling titanium alloy with a micro-texture ball-end milling cutter with uniform distribution density is proposed. When considering that the blunt edge radius of the ball-end milling cutter will cause the actual cutting radius to be smaller in the initial stage of milling [16], as shown in Figure 16c, it can be seen that

$$R_1 = R - \frac{r_\varepsilon}{\tan \theta} \quad (8)$$

In the formula, $R_1$ is the actual cutting radius, *R* is the theoretical cutting radius, $r_\varepsilon$ is the blunt radius of the cutting edge, and $\theta$ is the angle between the rake face and the center of the blunt circle.

According to Figure 15, the presence of the cutting edge will increase the value of *i*. Combined with Formula (3), the shear force and normal force on the shear surface will increase. With an increase in the radius of the blunt edge, the tool will become blunt and the milling force will increase. In addition, the presence of the edge will affect the distribution position of the micro-texture. At the same distance from the cutting edge, the presence of the blunt radius will cause the first row of micro-texture to be closer to the cutting edge, not only affecting the strength of the cutting edge, but also constraining the effect of the micro-texture. A schematic diagram is shown in Figure 17.

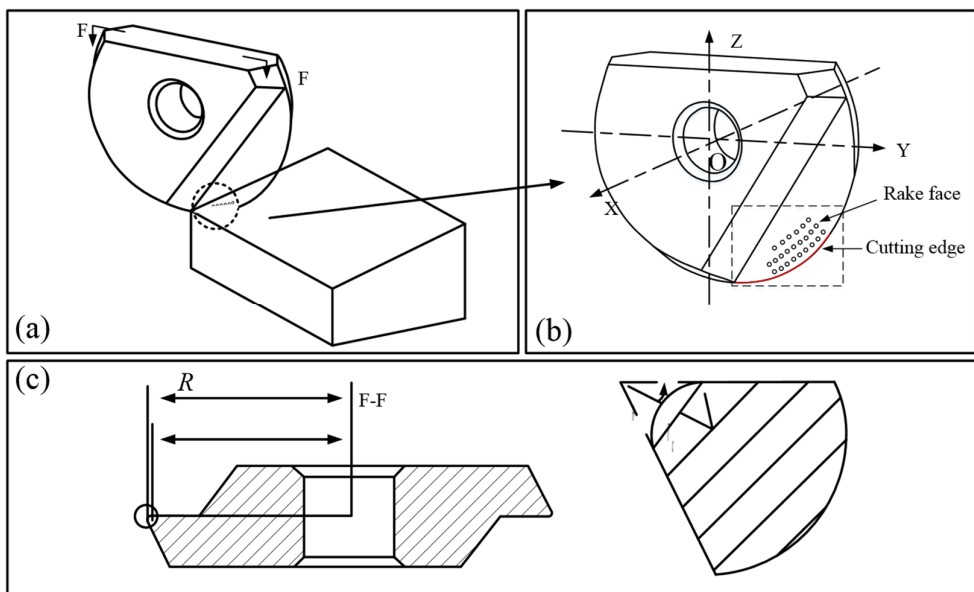

**Figure 16.** Model of titanium alloy milling with uniformly-distributed-density micro-texture ball-end milling cutter under the cutting edge: (**a**) A three-dimensional model of micro-textured ball end milling cutter milling titanium alloy under cutting edge; (**b**) micro-texture distribution on the rake face of ball end milling cutter; (**c**) blunt radius of ball end milling cutter edge.

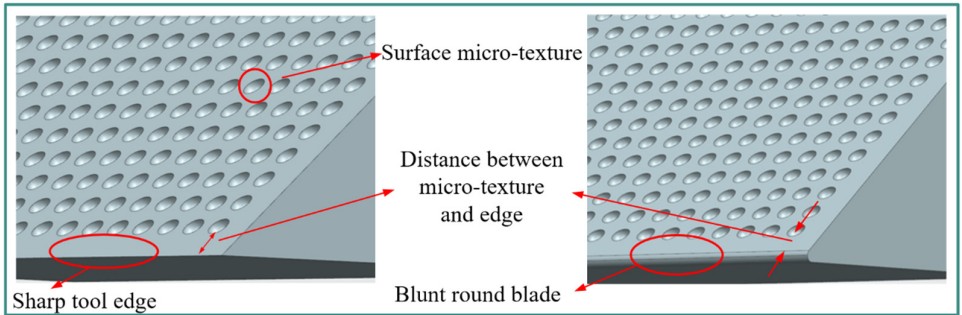

**Figure 17.** Relationship between the blunt edge and micro-texture distribution.

In summary, it is found that under different cutting parameters, the length and width of the tool–chip contact area will also be different, resulting in a change in the tool–chip contact area. Therefore, the actual tool–chip contact area with or without texture can be expressed as

$$\begin{cases} A_q(a_p, f, v) = l \cdot b \text{ (No micro-texture)} \\ A_q(a_p, f, v) = l \cdot b - n \cdot \pi r^2 \text{ (Micro-texture)} \end{cases} \tag{9}$$

In the formula, $n$ is the number of micro-textures in the tool–chip contact area, and $r$ is the radius of the micro-texture.

Analyzing Formulas (7) and (9), it can be found that the placement of micro-texture reduces the area of the tool–chip contact area, thereby reducing the friction F between the rake face and the chip and the normal force N of the rake face. From the above discussion, it can be seen that although the presence of the blade increases the milling force, it can improve the stability of the milling process; the micro-texture makes the contact surface between the tool and the chip smaller, which reduces the friction resistance, cutting force and cutting temperature. However, the blade radius of the blunt edge has a certain limit on the distribution position of the micro-texture and has a certain influence on the strength of the blade. Therefore, it is necessary to study the influence of the blunt edge parameters and micro-texture parameters on tool strength.

## 5. Strength Simulation of a Micro-Texture Ball-End Milling Cutter with Uniformly Distributed Density under the Action of the Cutting Edge

### 5.1. Experimental Design and the Establishment of the Finite Element Model

Considering the blunt edge parameters (blunt radius $r$), the micro-texture parameters (micro-texture diameter $d$, distance from the edge $l_1$ and micro-texture spacing $l$), and the interaction between the distance from the edge and the micro-texture diameter (expressed as $r \times l_1$), a four-factor three-level interaction test was designed. The design results are shown in Table 2. A control group comprising a ball-end milling cutter with no texture and no edge and only edge (20, 40, 60 µm) is added, and the numbers are set to 28, 29, 30, 31.

**Table 2.** Orthogonal test scheme for the strength simulation of a ball-end milling cutter after the edge and micro-texture are placed on the surface.

| Factor Test Number | $r$ (µm) | $l_1$ (µm) | $r \times l_1$ | $d$ (µm) | $L$ (µm) |
|---|---|---|---|---|---|
| 1 | 20 | 100 | 1 | 40 | 140 |
| 2 | 40 | 120 | 2 | 50 | 150 |
| 3 | 60 | 140 | 3 | 60 | 160 |

A three-dimensional model of the ball-end milling cutter is established in UG 10.0 (Unigraphics NX, Interactive CAD/CAM System). The tool material is cemented carbide, the front angle is 8°, the back angle is 11°, and the tool diameter is 20 mm. Considering the time required for the simulation calculation, only the blade part is simulated on the rake face of the ball-end milling cutter. It is also necessary to chamfer the cutting edge of the ball-end milling cutter. The three-dimensional model of the ball-end milling cutter is shown in Figure 18. Furthermore, the ANSYS 16.0 (computer-aided engineering (CAE) software, for performing finite element analysis) Workbench is used to simulate the force of the blade and the load condition of the micro-pit texture in the contact area of the ball-end milling cutter.

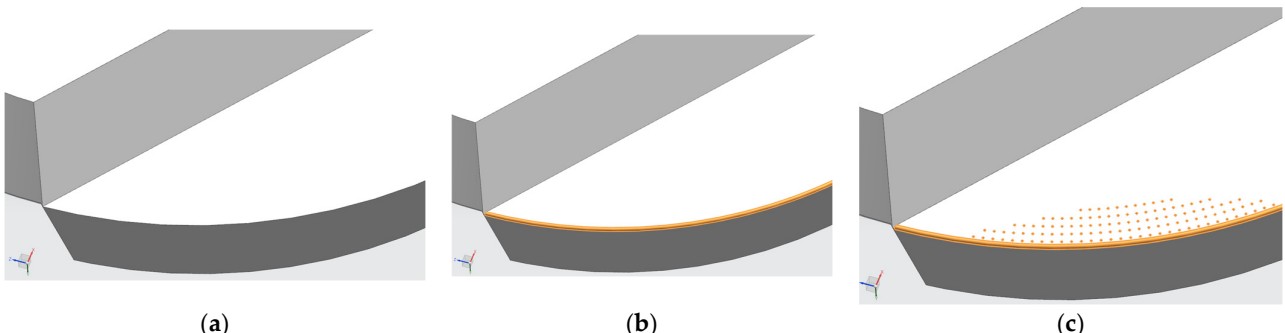

(**a**)      (**b**)      (**c**)

**Figure 18.** Three-dimensional model for a ball-end milling cutter. (**a**) Ball-end milling cutter; (**b**) blunt round-edge ball-end milling cutter; (**c**) micro-texture ball-end milling cutter with blunt round edge.

Because the material model in the ANSYS software material library contains no WC-Co cemented carbide (YG8) material, it needs to be individually set up. The density, elastic modulus, thermal expansion coefficient and Poisson's ratio of the material are manually input under the Engineering Data module. The material performance parameters of cemented carbide inserts are shown in Table 3.

**Table 3.** Cemented carbide tool performance.

| Chemical Composition | Density (g/cm$^3$) | Coefficient of Linear Expansion K (C$^{-1}$) | Young's Modulus (GPa) | Poisson Ratio | Tensile Yield Strength (MPa) |
|---|---|---|---|---|---|
| WC 92%, Co 8% | 14.5 | $4.7 \times 10^{-6}$ | 669 | 0.25 | 344 |

Meshing is an important step in ANSYS simulation. The overall and local meshing of the blade model is adopted, that is, the overall meshing parameters of the blade are 0.1 mm, and the meshing parameters of the tool–chip contact area are 0.05 mm [34]. The divided model is shown in Figure 19. The computational time is 0.004 s.

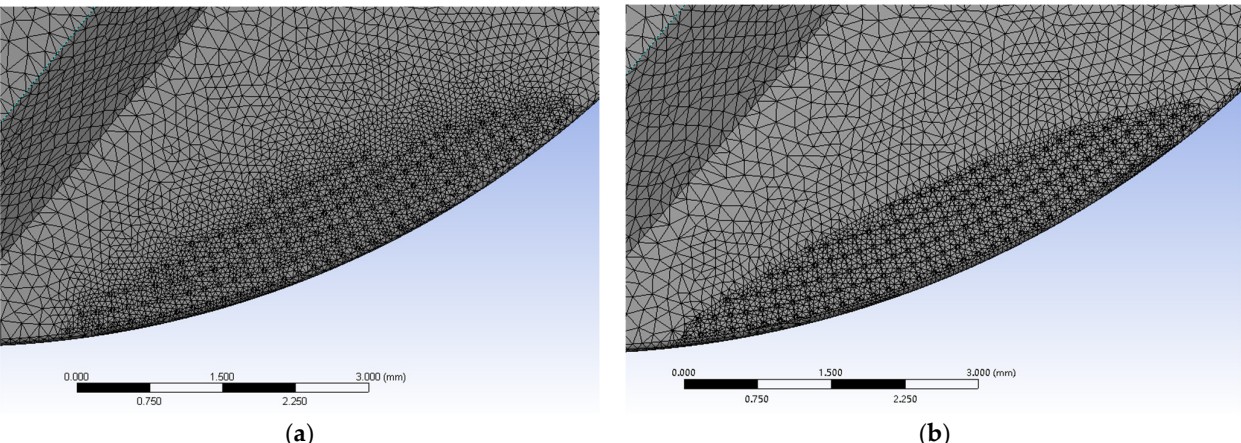

(a)                                                                                    (b)

**Figure 19.** Ball-end milling cutter model after meshing. (**a**) Unrefined grid; (**b**) refined grid.

Due to the static simulation performed using ANSYS, it is impossible to simulate the motion state of a workpiece milled using micro-texture tools. Therefore, a certain load is applied to the blade to simulate the working conditions. Considering the complexity of the force under actual working conditions, the force is simplified into three forms: one is to apply a concentrated load at the tooltip; the second is to apply linear load at the cutting edge position of the main and auxiliary cutting tools; the third is to apply surface load on the front and rear surfaces of the tool. The milling force is regarded as a linear surface load, which fluctuates in a sinusoidal function with time. The tool–chip contact area is the main area of cutting force and friction heat generation, so the micro-texture is placed in this position. The applied position of the load should also be the tool–chip contact area, that is, the area prepared for micro-texture, where a linear surface load is applied. In addition, constraints are set on the bottom of the tool, as shown in Figure 20.

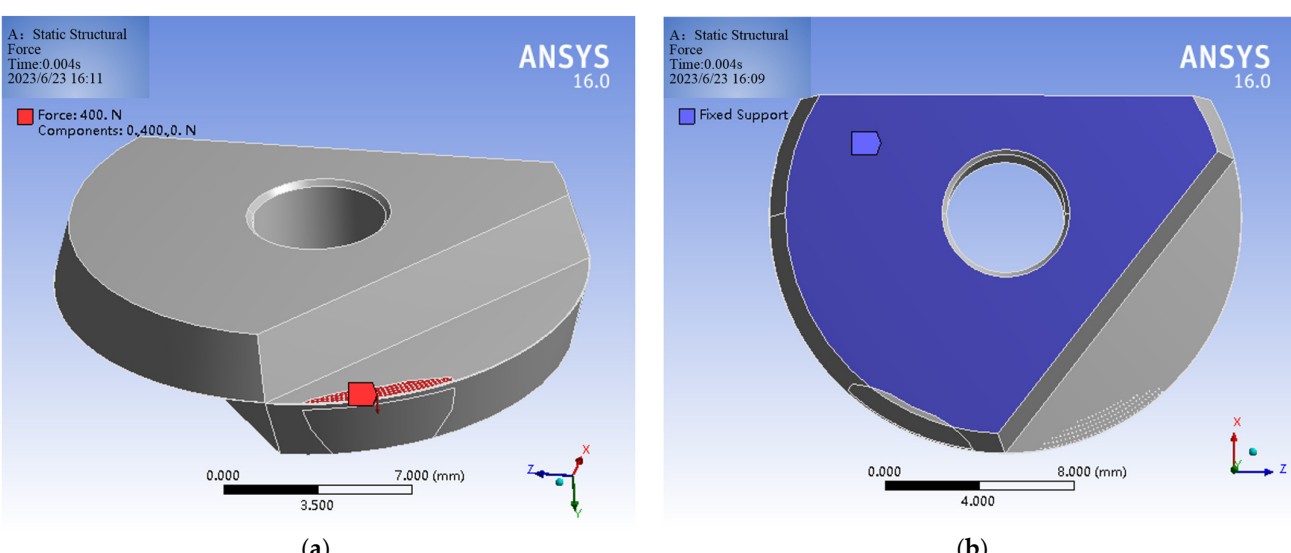

(a)                                                                                    (b)

**Figure 20.** Setting of simulation conditions. (**a**) Simulated boundary load; (**b**) constraint settings.

### 5.2. Analysis of Effect

The cemented carbide ball-end milling cutter can only bear limited deformation. When the external force on the blade increases to a critical value, the tool material will be damaged and broken. The transverse rupture strength (TRS) is used as the critical point to determine whether the tool has been broken. In the simulation analysis, the maximum principal stress is used to evaluate the fracture of the tool. When the maximum principal stress reaches the transverse fracture strength, TRS, of the tool material, the tool is considered to have been broken. The transverse fracture strength of WC-Co cemented carbide is about 2.5 GPa [35,36].

The stress and strain contours of the rake face after simulation are shown In Table 4. It can be seen from the deformation cloud diagram that the closer to the cutting edge of the tool, the greater the strain of the tool, and the maximum strain of the tool is found at the cutting edge. It can be found from the stress cloud diagrams that there is no stress concentration on the rake face of the tool, and the maximum stress value obtained in the 27 sets of tests is 233.51 MPa, which is far lower than the transverse fracture strength of the cemented carbide. It is proved that the insertion of the blunt edge and uniform-density micro-texture does not affect the normal use of the tool.

**Table 4.** Strain and stress cloud maps of the blunt-edge micro-textured ball-end milling cutter.

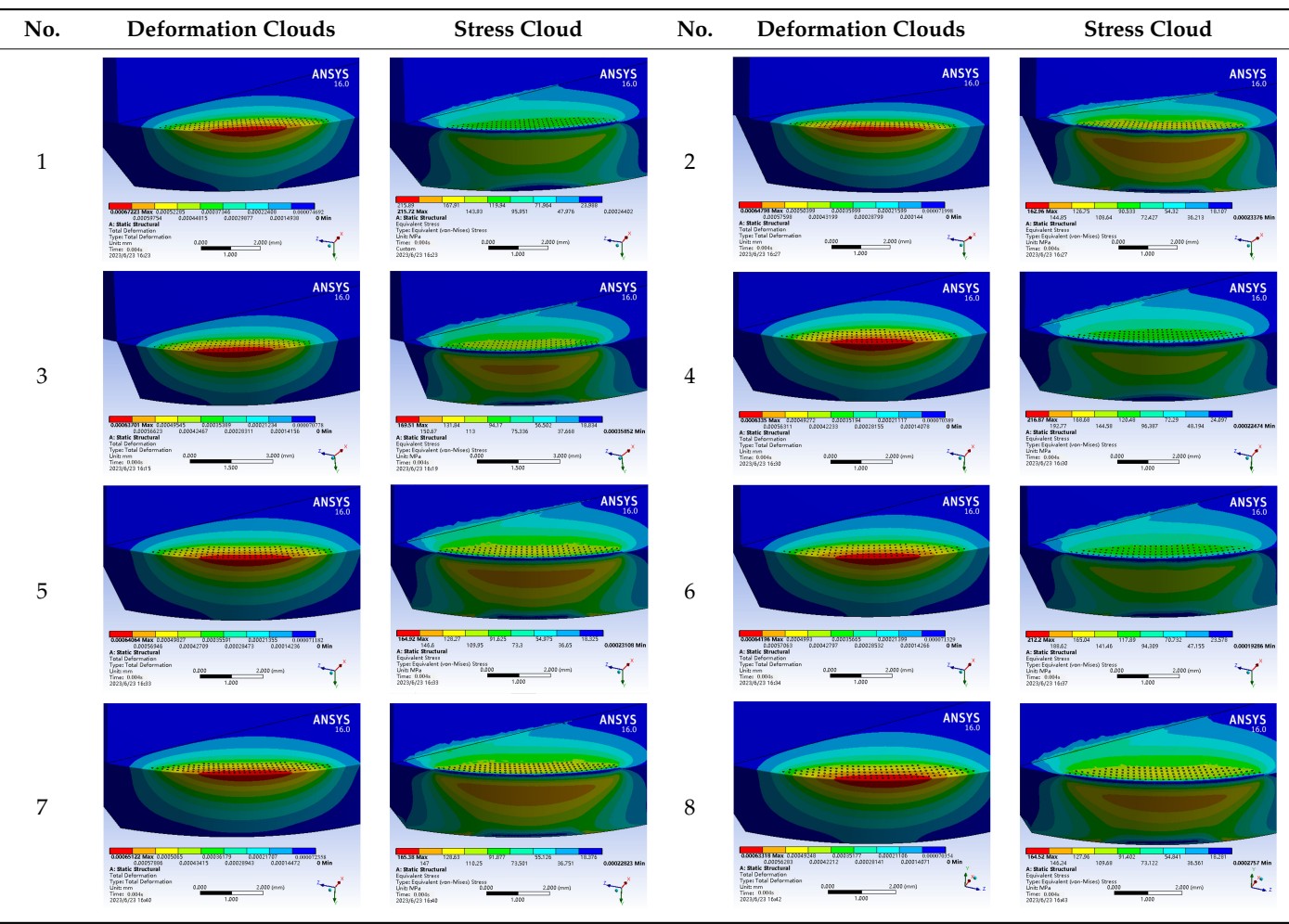

| No. | Deformation Clouds | Stress Cloud | No. | Deformation Clouds | Stress Cloud |
|-----|-------------------|--------------|-----|-------------------|--------------|
| 1   |                   |              | 2   |                   |              |
| 3   |                   |              | 4   |                   |              |
| 5   |                   |              | 6   |                   |              |
| 7   |                   |              | 8   |                   |              |

**Table 4.** *Cont.*

| No. | Deformation Clouds | Stress Cloud | No. | Deformation Clouds | Stress Cloud |
| --- | --- | --- | --- | --- | --- |

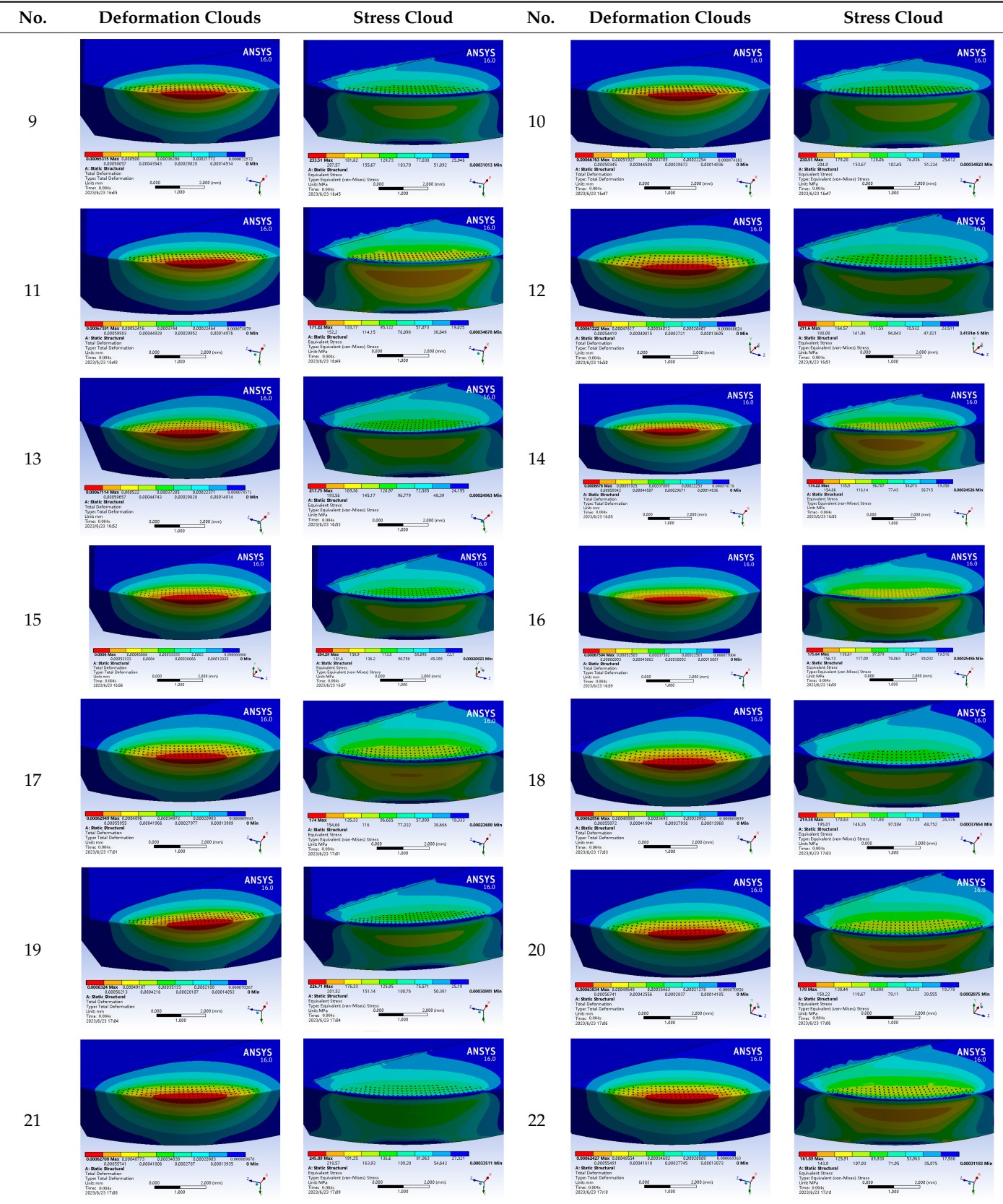

**Table 4.** *Cont.*

| No. | Deformation Clouds | Stress Cloud | No. | Deformation Clouds | Stress Cloud |
|-----|-------------------|--------------|-----|-------------------|--------------|

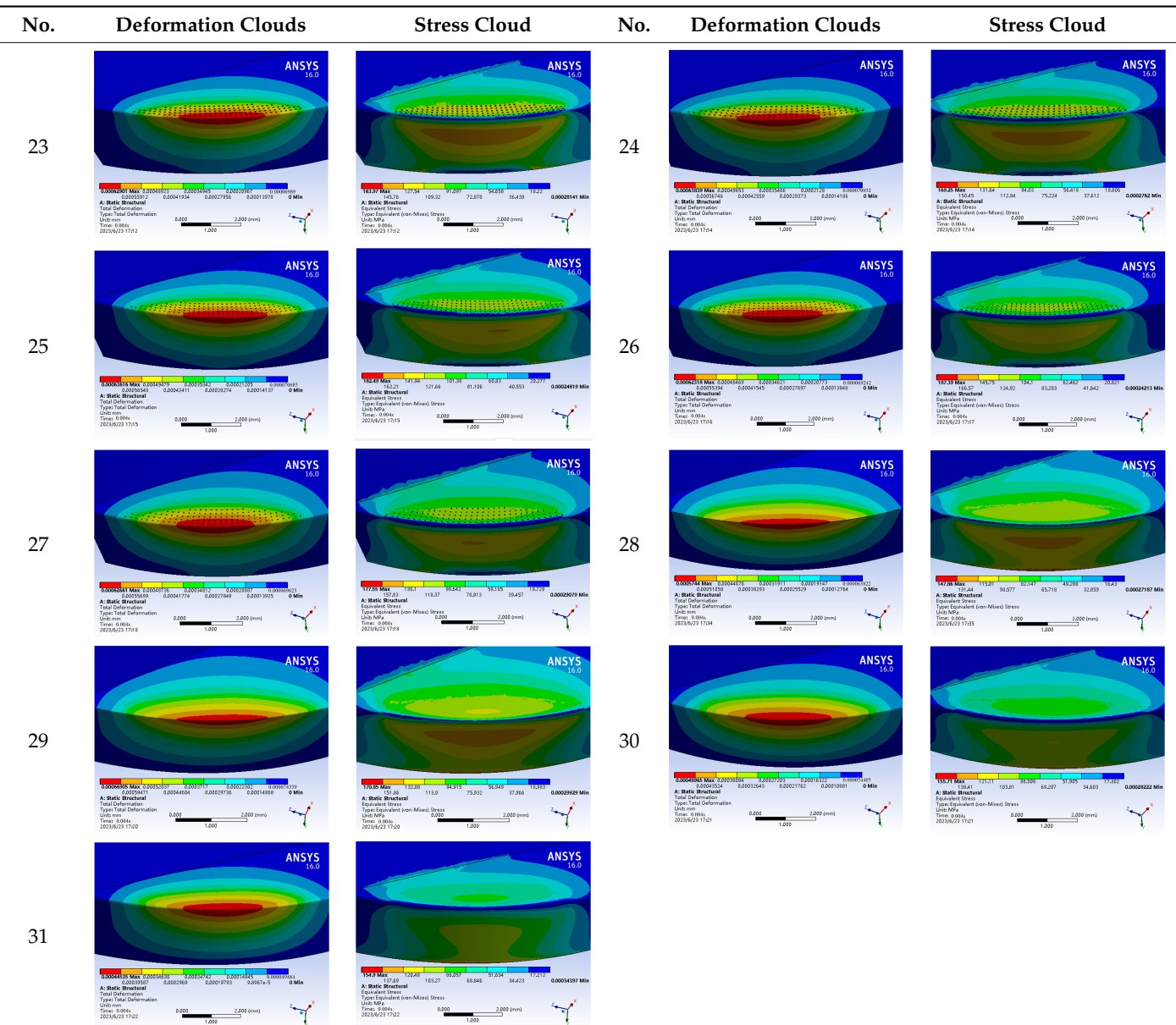

The maximum strain and stress of the non-edge non-textured tool are 0.0005744 and 147.86 MPa. The average maximum strain and stress of the cutting tool are only 0.000535 and 160.4867 MPa. The average maximum strain and stress of the micro-textured tool with a blunt round edge are 0.000643 and 192.2393 MPa. On the basis of a comparison between the non-textured non-edge tool and the edge tool, it is found that larger values of blunt edge radius can reduce the strain at the cutting edge of the tool, which is reduced by 6%. However, the maximum stress value increases by 8.5%. Furthermore, compared with the blunt-edge micro-texture ball-end milling cutter, it is found that the placement of micro-texture causes the maximum strain and stress to increase, with deformation increasing by 12% and 30%, respectively, and the stress increasing by 30% and 20%, respectively. This shows that under the same cutting conditions, the blunt-edge micro-textured tool will be damaged earlier.

The effects of blunt edge radius and micro-texture parameters on tool strength are analyzed using maximum strain and stress as evaluation indexes. On the basis of the data

presented in Table 4, a range analysis is carried out, and the results of the range analysis are shown in Tables 5 and 6.

**Table 5.** Analytical results of the maximum stress range of the micro-texture ball-end milling cutter with a blunt edge.

| Factor<br>Level | $r$ (µm) | $l_1$ (µm) | $r \times l_1$ | $D$ (µm) | $l$ (µm) |
|---|---|---|---|---|---|
| K1 | 189.5289 | 198.7411 | 179.1656 | 188.0056 | 199.2367 |
| K2 | 197.6233 | 187.2622 | 194.9722 | 198.8878 | 171.2444 |
| K3 | 185.5022 | 186.6511 | 198.5167 | 185.7611 | 202.1733 |
| R | 12.12111 | 12.09 | 19.35111 | 13.12667 | 30.92889 |
| rate | 4 | 5 | 2 | 3 | 1 |

According to Table 5, the main parameters affecting the maximum stress are the micro-texture spacing, the micro-texture diameter, and the interaction between the edge radius and the distance from the edge. The reason the micro-texture spacing and diameter affect the stress is that the change in the micro-texture spacing and diameter affect the distribution density of the micro-texture in the tool–chip contact area and change the area of the load. In addition, the distance from the blade and the radius of the blunt circle has little effect on the stress, but their interaction is obvious. This shows that the placement of micro-texture changes the stress distribution characteristics of the original blunt-edge tool.

The response surface methodology is used to analyze its influence law using Design Expert 13 software. Using the blunt radius, micro-texture diameter, distance from the edge, and micro-texture spacing as independent variables, and the stress value as the dependent variable, the response surface equation is established as follows:

$$
\begin{aligned}
F = {} & 166.11 - 4r - 3.78l_1 - 13.07D - 1.8l - 23.9r \cdot l_1 + 4.53r \cdot D - 1.59r \cdot l - 20.32l_1 \cdot D \\
& + 14.72l_1 \cdot l + 4.68D \cdot l + 7.7l_1{}^2 + 29.46l^2 - 10.87r \cdot l_1 \cdot l - 12.26r \cdot l^2 + 4.89r \cdot l_1{}^2 \cdot l
\end{aligned}
\tag{10}
$$

In the formula, $r$ is the blunt radius, $D$ is the micro-texture diameter, $l_1$ is the distance from the edge, and $l$ is the micro-texture spacing. Signature test: $R^2$ is 0.8942, $F$ statistic is 6.2, $p$ value is 0.0021.

According to the response surface diagram (Figure 21), it is known that the distance from the blade is 120 µm, acting as the dividing line. Below the dividing line, changes in the blade radius have little effect on the stress value, which only increases slightly with increasing blade radius. Above the dividing line, the stress changes significantly, and the stress value decreases with increasing edge radius. When the radius of the edge is 40 µm, the change in the distance from the edge has little effect on the stress value, and only decreases slightly with increasing distance from the edge. On the left of the dividing line, the stress changes significantly. As the distance from the edge increases, the stress value increases. It is found that when the edge radius is 60 µm and the distance from the edge is 140 µm, the maximum stress value is the lowest.

**Table 6.** Range analysis results of the maximum deformation of the micro-texture ball-end milling cutter with blunt edge.

| Factor<br>Level | $r$ (µm) | $l_1$ (µm) | $r \times l_1$ | $d$ (µm) | $l$ (µm) |
|---|---|---|---|---|---|
| K1 | 0.000645652 | 0.000645141 | 0.000642442 | 0.000639817 | 0.000648892 |
| K2 | 0.00064467 | 0.000635883 | 0.000639534 | 0.000638276 | 0.000642592 |
| K3 | 0.000630323 | 0.000639621 | 0.000638669 | 0.000642553 | 0.000629161 |
| R | $1.53289 \times 10^{-5}$ | $9.25778 \times 10^{-6}$ | $8.65556 \times 10^{-7}$ | $4.27778 \times 10^{-6}$ | $1.97311 \times 10^{-5}$ |
| rate | 2 | 3 | 5 | 4 | 1 |

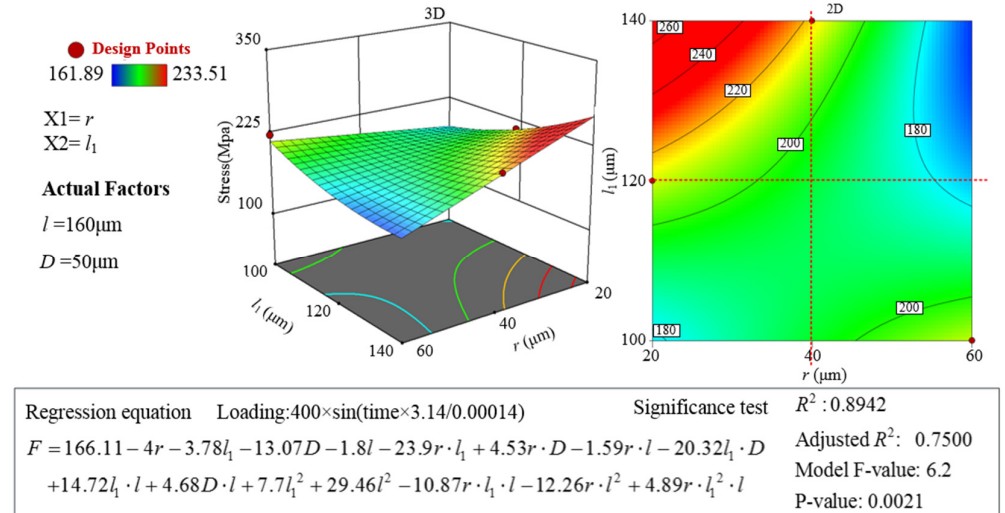

**Figure 21.** Response surface diagram of the interaction between the distance from the edge and the radius of the edge.

According to the range analysis results presented in Table 6, the change trend of tool deformation is not significant, and the values are distributed near 0.00063. The maximum deformation is 0.00067, and the minimum is 0.0006. In summary, the placement of blunt edges and micro-textures will have a certain impact on the stress and deformation of the tool. Their placement reduces the tool strength to a certain extent, and increases the cutting-edge deformation, but does not affect the normal use of the tool. In addition, the interaction of micro-texture spacing, edge radius and distance from edge greatly influences tool strength. Therefore, the micro-texture of the tool surface and the edge design need to be considered.

*5.3. Experimental Verification*

In order to verify the strength analysis of the blunt-edge micro-texture ball-end milling cutter described above, the 10th and 16th sets of tool parameters with the highest stress and strain were selected to complete the preparation of the blunt-edge and uniform-density micro-texture of the ball-end milling cutter. The results are shown in Figure 22. Cutting experiments are conducted using the prepared tool to verify the usability of the tool. The cutting parameters are selected as $a_p = 0.5$ mm, $f = 0.1$ mm/r, and $v = 140$ m/min. Milling force, tool wear, and surface roughness data are collected. A control group is added, and the experimental results are shown in Table 7.

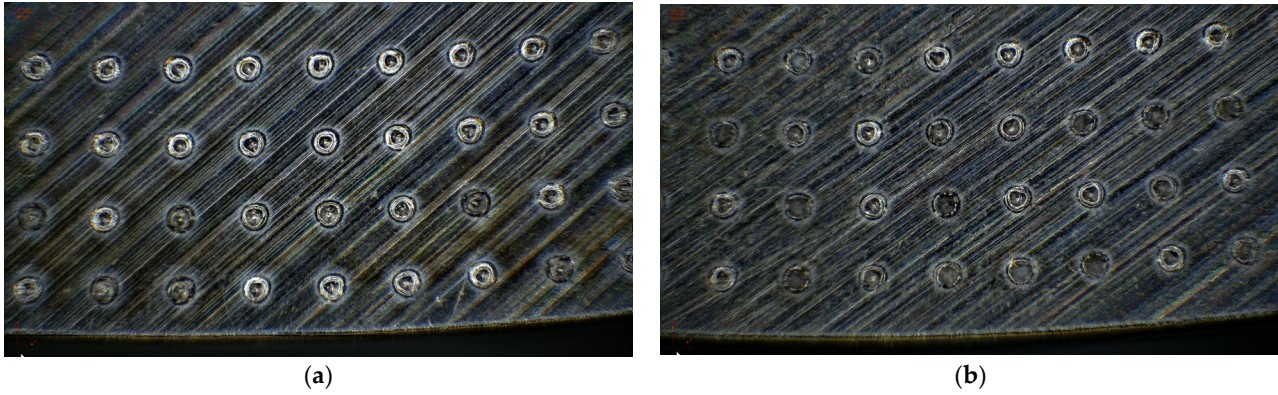

|     (a)     |     (b)     |

**Figure 22.** Preparation results of the blunt round-edge and micro-texture of the ball-end milling cutter. (**a**) Sample 10; (**b**) Sample 16.

**Table 7.** Range analysis results of maximum deformation of micro-texture ball-end milling cutter with blunt edge.

| Experimental Tool | Milling Force (N) | Tool Wear (µm) | Surface Roughness (µm) |
|---|---|---|---|
| 10 | 331.25 | 30.5 | 0.343 |
| 16 | 286.72 | 46.21 | 0.305 |
| no texture and no edge tool | 395.23 | 52.7 | 0.41 |
| 20 µm edge no texture tool | 342.73 | 50.45 | 0.366 |
| 40 µm edge no texture tool | 349.08 | 45.61 | 0.354 |
| 60 µm edge no texture tool | 363.51 | 47.33 | 0.352 |

Through experimental comparison, it can be found that the presence of the blunt edge and micro-texture does not affect the normal use of the tool, and the synergistic effect of the blunt edge and micro-texture has a positive effect on reducing milling force and tool wear, and improves the surface quality of the workpiece.

**6. Conclusions**

(1) Based on fractal theory, the tool morphology is studied before and after the actual machining, and the difference is defined. The tool–chip contact area is determined using the box dimensions. Compared with geometric analysis, this method is more accurate and effective.

(2) Based on the random point generation theorem of uniform distribution in the elliptical region, the distribution density function in the ball-end milling cutter region is generated, and its accuracy is determined by MATLAB simulation.

(3) It is proved that the micro-texture and blunt edge have a positive effect on the milling of the tool. However, the radius of the blunt edge restricts the distribution position of the micro-texture, thus affecting the strength of the tool.

(4) The placement of the micro-texture and the cutting edge has no effect on the normal use of the tool, but their placement reduces the strength of the tool to a certain extent and increases the deformation of the cutting edge.

There are still shortcomings in this study, and there is still much to be studied in the future. In order to facilitate the determination of micro-texture placement areas for other types of tools, a tool–chip contact area recognition platform based on fractal theory can be established. Furthermore, considering the different forms of tool–chip contact on the front face of the tool, a variable density micro-texture distribution function for ball-end milling cutters can be established. Finally, the performance of micro-textured coating tools under the action of the cutting edge needs to be studied.

**Author Contributions:** Conceptualization, S.Y. and P.H.; methodology, X.T. and X.W.; validation, X.T. and P.H.; formal analysis, P.H.; investigation, S.Y. and X.W.; resources, X.T. and X.W.; writing—original draft preparation, P.H.; writing—review and editing, S.Y.; funding acquisition, S.Y. and X.T. All authors have read and agreed to the published version of the manuscript.

**Funding:** This research was funded by the National Natural Science Foundation of China, grant no. 51875144 and 52005140.

**Institutional Review Board Statement:** Not applicable.

**Informed Consent Statement:** Not applicable.

**Data Availability Statement:** Not applicable.

**Conflicts of Interest:** The authors declare no conflict of interest.

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
