# Peer review of "Uniform-Density Micro-Textured Ball-End Milling Cutter Model Based on Fractal and Uniform Distribution Theory"

_coatings, doi:10.3390/coatings13081446_

Round 1
Reviewer 1 Report (New Reviewer)
The submitted paper deals with a method to determine the tool-workpiece contact area during milling and the effect of different micro-textured tools on their wear resistance. The paper is difficult to be read and the English usage is poor. The innovation of the submitted paper is questionable. Many research papers in the literature investigate the wear resistance of micro-textured tools. Moreover, many questions arise from the developed FEA model. For example, in the developed FEA model, three loading cases are examined. However, the authors have to show if the cutting loads distributions during milling resemble with any of the applied loads in the model, since they want to simulate the milling process. The figure 19 can’t be read, letters are too small. The authors have to describe the loading case corresponding to figure 19. Concerning the experimental part of the work, the authors have to describe how they produced the desired surface geometry. It’s compulsory for the authors to give some explanations why some surface geometries are better than others. Finally, the authors have to describe the type of wears developed during milling Titanium alloy. It was expected that due to the intense repetitive cutting loads in the cutting edge region near the flank during milling, fatigue is the main wear mechanism and the effect of friction on the wear of rake would not be so important. Moreover, another significant for the tool wear in milling titanium alloy is the thermal loads developed in the cutting edge region.
The english usage has to be improved.
Author Response
Please see the attachment.

Reviewer 2 Report (New Reviewer)
1. The subject addressed in the article is of current interest and, to a certain extent, consistent with the profile of the journal (the concepts "coated" and "coating" appear only in the list of bibliographic references).
2. The authors used modern research equipment and methods. However, the presentation of the authors' considerations is low.
3. It seems that the authors have published other works related to the article's subject proposed for publication, but without all such works being mentioned in the bibliographic reference lists. An example of such work is Tong Xin, Han Pei, and Yang Shucai. Coating and micro-texture techniques for cutting tools. J Mater Sci (2022) 57:17052–17104. Is there any reason why the mentioned work was not included in the list of bibliographic references?
4. It would be useful if some numerical values of the obtained results were entered in the final part of the Abstract (and in the Conclusions section).
5. The wording in the last part of the first paragraph of the introduction ("Generally, the tool needs to be passivated before use to improve tool availability and stability. Therefore, it is necessary to study the micro-textured tool under the action of cutting edge.") are confusing. It does not seem to be clear what the role of passivation is, nor why it can be deduced from the application of passivation that the mentioned study is necessary. A well-argued development of the passivation aspects could justify the paper's publication in the "Coatings" journal.
6. In line 42, it is mentioned that "The former proposes a new method...". The wording "an improved method ..." would correspond better to reality. The statement is also valid in the case of the formulation in line 87 ("a new idea").
7. Information of the type "based on AdvantEdge" or "based on Oxely" is incomplete since, for example, the concept "AdvantEdge" is also a stage venture capital fund based in Delhi (according to the information on the website https://www.advantedge.vc/. It is also necessary to avoid the inclusion of lesser-known abbreviations without being associated with the necessary explanations (see the case of the abbreviation "SPRT" in line 76, "UG" in line 54, etc.).
8. In the paper, different symbols are used for the same material (TC4, line 99, and Ti6Al4V, Table 2; I think the last symbol should be used, more frequently preferred in scientific works than the other symbol). In Table 2, it may be noted that the chemical composition is expressed in %.
9. The legend of Figure 1 is inadequate; there is no material information in the figure. The name "cutter handle" used in the figure is also inappropriate.
10. Why was the text in lines 137-159 highlighted?
11. Could a photo of the cutting area of the tool, showing how the insert is assembled to the body of the tool, be included in the article?
11. The authors believe that the contact area is highlighted by changing the appearance of the cutting tool surface. Is it possible that this area is larger, that is, it includes areas adjacent to the area considered by the authors of the article and in which, due to the lower pressure exerted by the chip, the high hardness of the tool material, and some elastic deformation of the surface layer of the tool, it did not lead to visible changes in the appearance of the tool surface?
12. Could some quantities be written along the abscissa and ordinate axes in the case of Figure 7? The request can also be made in the case of other diagrams included in the figures in the paper (for example, in Figure 12).
13. To whom do the two curves in Figure 8 correspond?
14. Formulation "In order to ensure that the micro-textured tool can give full play to the anti-wear and anti-friction effect when finishing titanium alloy, the micro-textured insertion area selects the union of the tool-chip contact area under different cutting parameters. The results of the tool-chip contact area are shown in Figure 10 a)” is confused. In the text of the article, other wordings can be considered confusing (for example, "According to Figure 13, the existence of the cutting edge will increase i." in line 309, "The text continues here." in line 353, "Considering the complexity of the tool force under actual working conditions,..." in line 378, etc.).
15. How was fractal theory used to develop the content of Figure 11?
16. It is necessary to introduce explanations for all symbols used in the algorithm described starting from line 254 (for example, there do not seem to be explanations for the symbols Rj, a, ηj in line 264). Some components of the sequence of instructions specific to the proposed algorithm are imperative; others are descriptive.
17. If the article's authors developed Figure 13, its content must be explained. What does the information "End face of processed material" mean in Figure 13?
18. Some arguments should be presented for using the hypothesis of uniform distribution density.
19. When, in line 329, the authors write that "Combining formula (4) and formula (6), ...", if it is a combination, a mathematical relation corresponding to the combination of the two formulas could be entered. However, it is possible that the authors had in mind the concept of "analyzing" instead of "combining".
20. At the current contrast level of the images in Figure 16, it is difficult to see the differences between the graphical representations, especially since there are no adequate explanations.
21. The information included in Table 3 is confusing. Not three tests were performed, as would appear from the first column in the table, but 27 tests, as can be concluded from Table 5.
22. What software was used to determine the regression equation in Figure 19 and to create the graph in this figure?
23. The "random point generation theorem" concept is mentioned only in conclusion no. 2. Where was it used in the article?
24. In preparing the list of bibliographic references, the recommendations in the instructions for authors were not followed (only the initials of the authors' first names were not used, the semicolon separator between the authors' names was not used, the expression "et al." in the case of the authors of a work is incomplete, symbols of type [J] are not necessary, etc.).
25. It is necessary for the authors to pay more attention to the editing of the paper and the expression in English
Thus, since all the authors work in the same place, it does not seem necessary to include the exponent "1" after each one's name or information regarding the authors' affiliation.
Generally, a blank space is left after a punctuation mark (see, in line 31, the placement of the parenthesis "(1)"). Other examples where blank space was not used where required are the following: ”[0.3,0.5mm],”, ”[120,160m/min]” in line 112, ”l1=0.11mm” and ”l1=0.15mm” in line 268, ” WC92%, Co8% ” in table 4, ” 0.1mm” in line 373, ” ap=0.5mm, f=0. 1mm/r, and v=140m/min.” in line 467, "Tool Wear(μm)" in Table 8, etc.
In line 125, two spelling marks appear after "Table 2".
The content of Figure 2 was included twice, with different magnifications. There are still figures where the images have been repeated (Figures 14, 15, etc.).
In separating the various components of the enumeration beginning in line 31, both the comma and the semicolon have been used. I believe that using the last spelling is more appropriate.
The regression equation in Figure 19 should be entered and commented on in the text of the article.
In lines 75, 78, etc., to exemplify the results of other researchers, some authors' first and last names are used, but it is recommended to use only the last name.
It is correct to write "WC-Co" instead of "WC-CO" in line 396.
See the comments for the authors.
Author Response
Please see the attachment.

Reviewer 3 Report (New Reviewer)
In this study, the ball-end milling cutter is taken as the research object, and the influence of micro-texture and edge on the tool is analyzed. The authors are advised to consider the following comments for this paper.
1. Which cutting tool material is represented by hard alloy? Detailed explanations are needed.
2. The authors said that “the milling method is down milling”. It should be discussed why it is selected to machine? I did not see any idea.
3. The authors said that “It can be seen that the curve equation results are in good agreement with the original results. The effectiveness of the analysis method is also proved.”. More detailed explanations with more quantitative data are needed.
4. Fig. 13b cannot be followed easily. It should be presented in a more understandable way.
5. No discussion about mesh dimension and computational time in finite element (FE) analysis is provided. They are important issues.
6. Kindly provide some recommendations for future studies in conclusion section.
7. I do not see some key references related to the research of the milling process.
Round 2
Reviewer 1 Report (New Reviewer)
The authors provided sufficient explanations for their work and the submitted paper can be accepted for publication. Please pay attention to the applied units for example strain is unitless (line 441). Only, in contours shown deformation can be used mm as units.
Author Response
Please see the attachment.

Reviewer 2 Report (New Reviewer)
In principle, the authors made corrections/modifications to the article's content, considering the opinions expressed by the reviewers.
However, some problems remain that can be solved in the final editing phase of the article.
Thus, in the list of bibliographic references, the authors maintained, in some cases, the wording "et al.", which seems to me at least impolite to the unmentioned co-authors (references no. 1, 6, 10, 11, 13, 14, 15, 17, etc.). In the case of reference no. 15, the information "(14pp)" is not necessary. In general, a blank space is left after a punctuation mark (see writing without such spaces in the case of reference no. 17 ("Ferroelectrics,2021,580(1),251-267"), but also in the case of other references.
See the comments and suggestions for the authors.
Author Response
Please see the attachment.

This manuscript is a resubmission of an earlier submission. The following is a list of the peer review reports and author responses from that submission.
Round 1
Reviewer 1 Report
My commentaries on the manuscript under consideration are as
follows:
1. The standards of the English are below those needed for
publication in international scientific journals.
2. The presentation is also far from that needed.
In particular, the experiments are not described in detail.
3. The key idea of the authors is to use the fractal theory in
order to describe the experimental data. The authors try
to determine formally the fractal dimension of the interface.
The fact that the object exhibits the fractal properties has,
however, not been demonstrated. From my point of view, there are
no any reasons to believe that the fractal theory is suitable here.
Physically sound models are here expected to be based on the data
of the interface structure and elasticity theory.
The authors do non mention the latter theory at all.
***
Taking the comments above into account, I believe that the
manuscript should be rejected.
See my report.
Reviewer 2 Report
The article presents a comprehensive study on the micro-textured ball-end milling cutter, focusing on its tool-chip contact area. The researchers establish a micro-texture distribution model using a plane rectangular coordinate system and conduct an experiment to simulate the forces at work on the blade. They analyze the effect of the cemented carbide ball-end milling cutter's ability to bear limited deformation, using the transverse rupture strength (TRS) as the critical point to determine tool breakage. The study concludes that the insertion of blunt edge and uniform density micro-texture does not affect the normal use of the tool, with the maximum stress value being far less than the transverse fracture strength of the cemented carbide. Minor revision is required.
-Improve Introduction section in terms of optimization studies. Some studies can be added below.
*Experimental investigation of newly designed cutting tool inserts in turning, Journal of Production Systems and Manufacturing Science
*Multi response optimization of turning operation with self-propelled rotary tool, Procedia-Social and Behavioral Sciences
*Development of an ANN-based decision-making method for determining optimum parameters in turning operation,Soft Computing
-The document does not provide explicit details on how these levels were specifically determined (in Table 1).
-Mechanical properties and chemical composition of Ti alloy should be given.
-Why was Ti alloy (TC4) used?
-Why was an orthogonal test plan used?
-Give more details about your FEM (boundary conditions, friction model, thermal properties, etc.)?
-In Table 4, the resolution of the figures should be improved.
-The article does not provide explicit details on how these experimental results were directly compared to the FEM results to validate them. The authors might have compared the experimental and FEM results visually or statistically, but such details are not provided.
The language and grammar used in the article seem to be appropriate. However, it's important to ensure that the article is free from grammatical errors and typos.
Reviewer 3 Report
The author includes more references in the introduction section. In the current version, only 15 references are cited.
The author includes the material properties of WC-Co cemented carbide in table form.
Which type of elements are used for simulation?
Mention the boundary conditions for the ball end milling cutter process.
Table 4 is not unclear. What do those terms in the header stand for?
The author includes a regression equation for comparing the test results and also mentions the error percentage of the strain rate.
The author mentions a comparative statement of different processes or previous work done by the researcher related to the ball-end milling cutter process.
Once the author carries out all the corrections, the paper will be accepted.
In the current version, a major revision is required.
Round 2
Reviewer 1 Report
My comment was that there are no any reasons to believe
that the fractal theory is suitable for the system under
consideration. Here, I can add that this theory implies
self-similarity of the interface. In the response,
the authors outline some elements of the fractal theory
and mention self-similarity, but it was not proven
in their work. For this reason, the main concept of
their work and the emphasis on the fractal theory are
misleading, and accordingly I repeat my recommendation to reject the manuscript.
See my first report.
Reviewer 3 Report
Accepted in the present form.
Accepted in the present form.